# Combinatorial Optimization via Memory Metropolis: Template Networks for Proposal Distributions in Simulated Annealing applied to Nanophotonic Inverse Design

## Abstract

We propose to utilize a neural network to build transition proposal distributions in simulated annealing (SA), which we use for combinatorial optimization on 2D-binary grids and thereby direct convergence towards states of structurally clustered patterns. To accomplish this we introduce a novel class of network architectures called template networks. A template network learns a template to construct a proposal distribution for state transitions of the stochastic process of the Metropolis algorithm, which forms the basis of SA. Each network represents a single constant pattern and is trained on the evaluation results of intermediate states of a single optimization run, resulting in an architecture not requiring an input layer. Using this learning scheme we equip the Metropolis algorithm with the ability to utilize information about past states, intentionally violating the Markov property of memorylessness, and therefore call our method Memory Metropolis (MeMe). Moreover, the emergence of structural clusters is encouraged by incorporating layers with limited local connectivity in the template network, while the network depth controls the learnable cluster sizes. Viewing the optimization objective of the Metropolis algorithm as a reward maximization allows to train the template network to find high-reward template-patterns.

We apply our algorithm to combinatorial optimization in nanophotonic inverse design and demonstrate that MeMe results in clustered design patterns suitable for direct optical chip fabrication which can not be found by plain SA or regularized SA. Code is available at https://XXXXXXXX.

## 1 Introduction

Finding specifically constrained patterns on a binary grid is a challenging combinatorial optimization problem with prominent applications in nanophotonic inverse design (Molesky et al., 2018; So et al., 2020; Piggott et al., 2017; Liu et al., 2018), especially when building compact and high-performance devices for nanophotonic integrated circuits (Moody et al., 2022). A nanophotonic device represents a single functional building block which performs arbitrary operations on light propagating through a photonic circuit, ranging from simple (e.g. equally distributing light to multiple outputs) to highly complex (e.g., efficient demultiplexing of information encoded in different wavelengths). The goal of nanophotonic inverse design algorithms is to optimize the material distribution on an optical chip in a bounded design area in between input and output waveguides. We map the design area to a 2D-grid and focus on binary material distributions, i.e., at each grid cell material may either be present or not, thus leading to a binary combinatorial optimization problem (see Fig. 1A). While the efficiency of a device may directly be obtained from the electromagnetic field distribution for a certain input, which can be computed with high accuracy through finite-difference frequency-domain (FDFD) simulations, the fabricability of the evaluated material distribution pattern (called state $S$ in the following) must also be taken into account. Isolated pixels representing single pillars or holes of very small diameters may not be feasible for established fabrication processes. Furthermore, larger clustered regions with less material edges are reducing scattering and thus signal loss (Hughes et al., 2005) and are further easing fabrication. Therefore, an optimal algorithmic implementation yields both, highly efficient and locally clustered solutions.

We seek to solve the combinatorial optimization problem of finding clustered patterns on a binary grid to achieve a desired photonic functionality which we evaluate using FDFD-simulations. This is searching for the optimal (clustered) state

$$\boldsymbol{S^*} = \underset{\boldsymbol{S} \in \{-1,1\}^{d_x \times d_y}}{\arg\max} \mathcal{F}(\boldsymbol{S}) \tag{1}$$

with grid pixel values -1/1 representing material unfilled/filled cells, the fitness function $\mathcal{F} : \{-1, 1\}^{d_x, d_y} \to [0, 1]$ defining the desired functionality and incorporating the FDFD-simulations and design area dimensions $d_x, d_y$. As a baseline algorithm we consider this problem as a Markov chain Monte Carlo (MCMC) problem and search for optimal states by Simulated Annealing (SA) (Kirkpatrick et al., 1983) based on the Metropolis algorithm (Metropolis et al., 1953) (i.e., Metropolis-Hastings algorithm for Boltzmann distributions; Hastings (1970)). We select next state proposals by randomly (uniformly distributed) inverting single pixels and use negative fitness values ($f_t = F(S_t) = -E(S_t)$) as energies, resulting in pixel inversions being accepted with probability $p = \min(1, \exp((f_{t+1} - f_t)/T_M^t))$, with decaying Metropolis temperature $T_M^t$.

However, we identify multiple problems with this approach which are tackled by our proposed algorithm. We observe regions where unfilled pixels are highly detrimental for the device (e.g., in front of waveguides). The random selection of pixel inversion proposals results in computationally heavy FDFD-simulations required to evaluate inverting those pixels, even though these inversions will rarely be accepted. Likewise, we observe beneficial regions which, however, are explored slowly. Reaching a final state where these are fully exploited demands a very slow annealing of $T_M$, resulting in many steps needed. Moreover and most importantly, converged states do not form densely connected clusters and are thus not suited for actual fabrication.

We solve these problems by proposing pixel inversion candidates with a specially crafted class of deep neural networks we call template networks, which are trained to learn a single pattern. By training the network information about previously evaluated pixel inversions is stored and propagated to patterns in a local vicinity by using locally connected layers. After appropriate normalization we sample inversion proposals from the discrepancy between the template network output and the current state and evaluate the Metropolis criterion to construct a stochastic process of states $\{\boldsymbol{S}_t\}$. Since the network accumulates information about past visited states, we break with the Markov property of being memoryless. Thus, we call our proposed method Memory Metropolis (MeMe).

Despite the so constructed proposal distribution not being symmetric, we do not consider the transition probabilities when evaluating the Metropolis criterion. We therefore bias the state distribution and the resulting stochastic process does not follow the underlying Boltzmann distribution of fitness values anymore. Instead, the transition distribution is shifted towards the proposal distribution, resulting in a state distribution which is a combination of the Boltzmann distribution of fitness values and the learned template pattern, which encourages cluster formation.

To train the network we consider MeMe as a reinforcement learning (RL) algorithm, where actions are pixel inversions, the policy is learned by the template network and differences between fitness values of consecutive states are interpreted as rewards. Similar to deep Q-learning, the network output is interpreted as a reward prediction and used to generate a distribution for action sampling. However, to learn a fixed template, we directly learn single reward differences without considering future Q-value contributions, i.e., Q-learning with maximal discount ($\gamma = 0$). Instead of a temporal difference loss (e.g. squared error), we utilize a loss based on the product of network predictions and rewards to additionally incorporate network certainty into the predictions to calculate policy gradients. The Metropolis criterion allows the agent to directly revert detrimental actions without the need to first update the policy or to reevaluate the environment. In summary our contributions are:

1. We propose to solve combinatorial optimization problems of nanophotonic inverse design by extending simulated annealing (SA) to form a reinforcement learning agent.

2. We propose to sample state transitions of SA from a learned neural network storing information about past states (Memory Metropolis; MeMe).

3. We introduce template networks, a novel neural network architecture which learns structured patterns due to a combination of backward and forward propagation in the network, despite no input layer or activations being used.

4. We evaluate our algorithm extensively in the context of inverse nanophotonic designs and demonstrate that our approach can generate results beyond the capabilities of SA opening up new experimental possibilities.

In the following we focus on the description and analysis of the combinatorial optimization algorithm, the deep learning architecture and the reinforcement learning framework. More details on the application of our algorithm in the context of nanophotonics will be presented in an additional publication which is prepared in parallel to this work.

## 2 RELATED WORK

Reinforcement Learning (RL) is applied to diverse combinatorial optimization problems (Bengio et al., 2021; Mazyavkina et al., 2021). Some past works also propose combinations of RL and SA or the Metropolis algorithm, however, none of these use deep neural networks for proposal generation of discrete states, deliberately bias the state distribution of SA or add the Metropolis criterion to an RL agent to revert actions.

Guo et al. (2004) are utilizing the Metropolis criterion in Q-learning to choose between an action from the agents policy or a random action based on Q-values as energies to achieve additional exploration. Szewczyk & Hajela (1993) are using a neural network to predict energies for SA in combinatorial optimization. A RL-agent to learn the temperature scheduling of SA was proposed by Mills et al. (2020). Similarly, other works use RL on top of heuristic optimization algorithms to control its hyperprameters (Beloborodov et al., 2021; Khairy et al., 2020; Wauters et al., 2020). Proposal distributions of SA where learned via supervised learning on stored trajectories by Alvarez et al. (2012). Similar to MeMe, Correia et al. (2023) combine deep RL and SA, however, they do not use the learned proposal distribution to bias the optimization to converge to a constrained state, they rely on hand-crafted features as inputs to FC layers where we propose input-less template networks and update the network predictions by evolution strategies. For the Metropolis-Hastings algorithm (Hastings, 1970) leapfrog integration of Hamiltonian dynamics Duane et al. (1987); Neal (2011) can be used for proposal generation. (Levy et al., 2018) propose to learn the leapfrog operator based on neural networks, which was further adapted by Li et al. (2021) and Hoffman et al. (2019) among others. Similarly, Müller et al. (2019) use a neural network for minimizing estimation variance in Monte Carlo integration. Xia et al. (2022) are extracting a proposal distribution for Metropolis-Hastings from a LSTM trained on a separate training dataset and apply their algorithm to outlier detection.

Related to our template networks, grids of learned parameters without explicit input layers followed by fully connected layers are used in neural scene rendering from 2D-images (Mildenhall et al., 2020; Sun et al., 2022). Similar to our approach, these networks are fully fitted to a single scene. However, the applied network architectures do not exploit local connectivity to learn local patterns. Multiresolution hash encodings Müller et al. (2022) offer the ability to share information among grid cells due to intended hash conflicts.

Locally connected layers where utilized by e.g. Gregor & LeCun (2010) and Huang et al. (2012).

Machine Learning, and especially RL, was applied to nanophotonic inverse design by Li et al. (2023); Moody et al. (2022) and Dinsdale et al. (2021).

Commonly employed approaches for nanophotonic inverse design cover gradient based methods (Piggott et al., 2014; 2015) often combined with level-set functions Piggott et al. (2017) to transform continuous solutions to discrete spaces, direct binary search trees Jia et al. (2018); Shen et al. (2015) and genetic algorithms Spuhler et al. (1998). Apart from nanophotonic inverse design, RL was also applied to find patterns in e.g. nuclear assembly (Radaideh et al., 2021) or molecule design (Zhou et al., 2019).

## 3 METHODS

In the following we describe MeMe and template networks in the framework of reinforcement learning (RL). We focus on 2D binary grids, however, applications to other search spaces can be constructed analogously. We denote 2D matrices representing quantities of the grid cells as bold capital symbols (e.g. $\boldsymbol{Q}_t^{ij}$) where upper indices represent spacial coordinates and lower indices represent time steps.

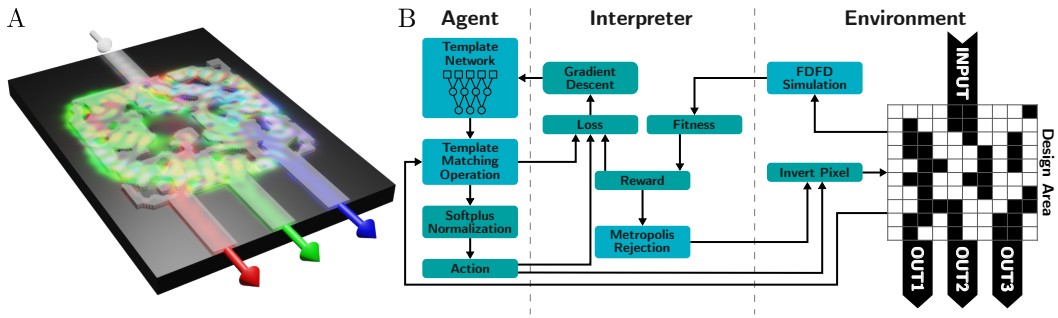

Figure 1: **A**: Nanophotonic device for optical processing with optimized binary pattern implementing a wavelength demultiplexer overlayed with the electromagnetic energy density (false-colored). **B**: Schematic illustration of MeMe viewed from the perspective of RL.

---

**Algorithm 3.1:** Memory Metropolis (MeMe)

---

**Input:** Design area dimensions $d_x$ and $d_y$, sampling temperature $T_S$, metropolis base
temperature $T_M^0$, learning rate $\eta = 1$, iterations $t_{max}$

**Initialize:** State $\boldsymbol{S}_0 \in \{-1,1\}^{d_x \times d_y}$ here: $\boldsymbol{S}_0^{i,j} = 1 \, \forall i,j$; network parameters $\Theta_0$

**for** $t \leftarrow 0$ **to** $t_{max}$ **do**

 $\boldsymbol{N}_t = \mathcal{N}(\Theta_t)$       // evaluate template network

 $\boldsymbol{Q}_t = -\boldsymbol{N}_t \odot \boldsymbol{S}_t$      // template matching operation (TMO)

 $\boldsymbol{P}_t^{ij} = \log(1 + \exp(\boldsymbol{Q}_t^{ij}/T_S))/\sum_{i,j} \log(1 + \exp(\boldsymbol{Q}_t^{ij}/T_S))$   // normalization

 $a_t = (a_t^x, a_t^y) \sim \boldsymbol{P}_t$      // sample action (pixel indices)

 $\tilde{\boldsymbol{S}}_t = \boldsymbol{S}_t$         // copy state

 $\tilde{\boldsymbol{S}}_t^{a_t^x, a_t^y} = -1 \cdot \tilde{\boldsymbol{S}}_t^{a_t^x, a_t^y}$      // invert pixel

 $\tilde{f}_t \leftarrow \mathcal{F}(\tilde{\boldsymbol{S}}_t)$    // evaluate fitness function (FDFD simulation)

 $r_t = \tilde{f}_t - f_t$        // calculate reward

 $T_M^t = T_M^0(1 + \cos(\pi t/t_{max}))/2$     // simulated annealing

 **if** $\exp(r_t/T_M^t) < X \sim \mathcal{U}(0,1)$ **then**    // metropolis criterion

  $\boldsymbol{S}_{t+1} = \boldsymbol{S}_t$        // revert action

  $f_{t+1} = f_t$

 **else**

  $\boldsymbol{S}_{t+1} = \tilde{\boldsymbol{S}}_t$       // accept action

  $f_{t+1} = \tilde{f}_t$

 **end**

 $\Theta_{t+1} = \Theta_{t+1} - \eta \partial \mathcal{L}(\boldsymbol{Q}_t, r_t, a_t))/\partial \Theta_t$    // policy learning via SGD

**end**

**return** $\boldsymbol{S}_{t_{max}}$

---

### 3.1 MEMORY METROPOLIS

Fig. 1B schematically depicts MeMe viewed from the perspective of RL while a detailed description is given in Alg. 3.1 a detailed listing of our notation is given in Appx. A.16. Starting from an all-ones (all-material) state we consecutively update states by inverting single pixels.

**Action Proposal:** For every iteration the template $\boldsymbol{N}_t \in \mathbb{R}^{d_x \times d_y}$ is calculated as the output of the template network $\mathcal{N}(\Theta_t)$, followed by comparing the template with the current state $\boldsymbol{S}_t$, which we call template matching operation (TMO). Since the states are defined on a binary grid, $\boldsymbol{S}_t^{ij} \in \{-1,1\}$, this can be done by multiplying the state elementwise with the learned template. The resulting values $\boldsymbol{Q}_t = -\boldsymbol{N}_t \odot \boldsymbol{S}_t$ represent the dissimilarity between template and state as well as the reward expected from inverting the corresponding pixels (see Sec. 3.2 for a more detailed discussion of template networks).

Next $\boldsymbol{Q}_t$ is normalized to derive the proposal distribution (action policy) $\boldsymbol{P}_t \in [0,1]^{d_x \times d_y}$. We first divide $\boldsymbol{Q}_t$ by the sampling temperature $T_S$, allowing to control the amount of random exploration.

While low temperatures make the agent strictly follow the networks predictions, higher temperatures result in increasingly uniformly sampled actions. The scaling is followed by a normalized softplus function, i.e., element-wise applying $f(x) = \log(1 + \exp(x))$ and normalizing the sum over all elements to 1. We chose to use the softplus function instead of an exponential as done in the broadly used softmax function since softplus approximates a linear function for high inputs. Thereby, the exploration-exploitation trade-off is independent of the network output scale for large output magnitudes.

**State Evaluation:** After sampling an action $a_t$ (i.e., pixel indices), a candidate for the next state $\tilde{\boldsymbol{S}}_t$ is constructed by copying the current state and inverting the pixel corresponding to the action. Next the fitness of the candidate state $\tilde{f}_t = \mathcal{F}(\tilde{\boldsymbol{S}}_t)$ is evaluated. The definition of $\mathcal{F}$ depends on the desired nanophotonic functionality, however, evaluating $\mathcal{F}$ always requires solving FDFD simulations which are computationally costly and thus mainly determine the total algorithms runtime. The reward is calculated as the change of fitness.

**Metropolis criterion:** Additionally to standard RL agents, we apply the Metropolis criterion to revert detrimental actions, i.e., an action is reverted with probability $\min(0, 1 - \exp(r_t/T_M))$. We anneal the Metropolis temperature $T_M$ following a half-period cosine annealing schedule (simulated annealing; SA) to converge to the states of high fitness. The Metropolis criterion allows MeMe to directly revert detrimental actions without the need to reevaluate the state and sample a new action and thus reduces computationally costly FDFD-simulations.

**Policy/template network learning:** We train the network by a simple policy gradient algorithm. We chose to minimize the product of the TMO output and the reward, i.e., $\tilde{\mathcal{L}}(\boldsymbol{Q}_t, r_t, a_t) = -r_t \boldsymbol{Q}_t^{a_t^x, a_t^y}$, instead e.g. a loss based on the temporal difference (i.e. squared error). This loss function trains the network to predict whether an action will cause a negative or positive reward while the magnitude of the prediction represents a combination of the magnitude of the expected reward and the certainty of the predictions. Pixels which have been sampled multiple times and caused rewards of the same sign will cause $\boldsymbol{Q}_t^{a_t^x, a_t^y}$ to grow continuously. In combination with the TMO this causes those actions to be sampled less. On the other hand, actions which have rarely been sampled or which yielded ambiguous rewards will be sampled with higher probability. Directly learning to predict the reward (e.g. by a squared error loss) causes low-importance pixels (i.e. reward close to zero) to be inverted back and forth endlessly, resulting in non-clustered regions. Furthermore, the continuously increasing loss function allows for an inherent exploration-exploitation adaption during training similar to the annealing schedule we apply to $T_M$. To accomplish fast learning for very beneficial/detrimental actions while still retaining training stability, we employ a dampened version of the product loss:

$$\mathcal{L}(\boldsymbol{Q}_t, r_t, a_t) = -\text{sign}(r_t \boldsymbol{Q}_t^{a_t^x, a_t^y}) \log(1 + |r_t \boldsymbol{Q}_t^{a_t^x, a_t^y}|). \quad (2)$$

Network parameters are updated by performing a gradient descent update step. We neither use momentum nor adaptive optimizers since completed actions should impact the network immediately. Since we initialize the template network to have an all-zero output, increasing the learning rate $\eta$ has a similar effect to decreasing $T_S$, thus we keep $\eta = 1$ fixed and only tune $T_S$ if needed.

**Markov property:** Compared to the Metropolis algorithm we added a proposal distribution and deliberately did not take the asymmetry of the proposal distribution in the evaluation of the metropolis criterion into account (cf. Appx. Algo. A.1), which results in violating the Markov property and biasing the transition distribution from a Boltzmann distribution of the fitness differences towards the proposal distribution (see Appx. A.15 for exact derivation and Appx. A.9 for results for unbiased sampling). This results in converging to states that consists of clustered patterns, since the proposal distribution learned form the template network also forms clustered patterns. The balance between both distributions is determined by $T_S$ and $T_M$ (cf. Appx. Fig. A.10). $T_S = \infty$ results in a uniform proposal distribution, reducing MeMe to plain SA (cf. Appx. Algo. A.4) and thus, states are following the Boltzmann distribution again. $T_M = \infty$ causes the Boltzmann distribution to be replaced by a uniform distribution, so states will fully follow the proposal distribution and no actions will be reverted. Since the proposal distribution (as policy of an RL agent) is trained to maximize the reward, the latter case also converges to high fitness states. However, combinations of finite $T_S$ and $T_M$ allow tuning tradeoffs between exploration and exploitation and cluster formation and fitness maximization.

## 3.2 TEMPLATE NETWORKS

The template network is designed to learn a single pattern, thus no input layer is needed. Instead, the first layer of the template network consists of a grid of learned parameters, followed by multiple

2D-locally connected layers. Locally connected layers are connected in the same way as convolutional layers but do not share weights among spatial dimensions (cf. Fig. 2A). The locally connected layers enable the network to learn local relations and thus encourage cluster formation. Since there is no input, the output is constant and could also be directly extracted from a simple table, however, the learning dynamics would differ significantly from our proposed architecture. When backpropagating gradients through the network during the backward path, the locally connected layers propagate the gradient to a broader neighborhood with increasing network depth. The networks subsequent forward pass distributes the updated weights up to a distance of $dk/2$ where $d$ is the networks depth and $k$ the connectivity (cf. filter size in CNNs). Fixing all weights of the locally connected layers to the same value is equal to convolving the first learned parameter layer with a binomial (or Gaussian) kernel (cf. Fig. 2B). We initialize the first learned parameter layer to zeros and all locally connected layers weights to $1/k^2$. Thus, Gaussian-like cluster formation is encouraged by the network output. However, since weights will change during training and are not shared across spatial dimensions (as in convolutional layers), activations can decouple and cluster formation is not enforced strictly. This decoupling especially takes place if neighboring neurons are backpropagating contradicting gradients, hence connecting weights will be reduced resulting in dismantling the local connectivity. This allows for sharp-edged structures to be learned, if beneficial for the optimization objective (cf. Appx. Fig. A.1). However, regions of low importance (low reward and thus low gradients) form dense structures. Furthermore, the network architecture allows incorporating prior knowledge similar to importance sampling in Metropolis-Hastings by setting the initialization values of the learned parameter layers. E.g. connecting lines between input and output waveguides can significantly speed up optimizations (cf. Appx. A.6). Hard constraints can be included by additionally freezing or masking layers. Lastly, the template network output is compared against the current state to construct inversion probabilities (template matching operation; TMO; see Fig. 2C).

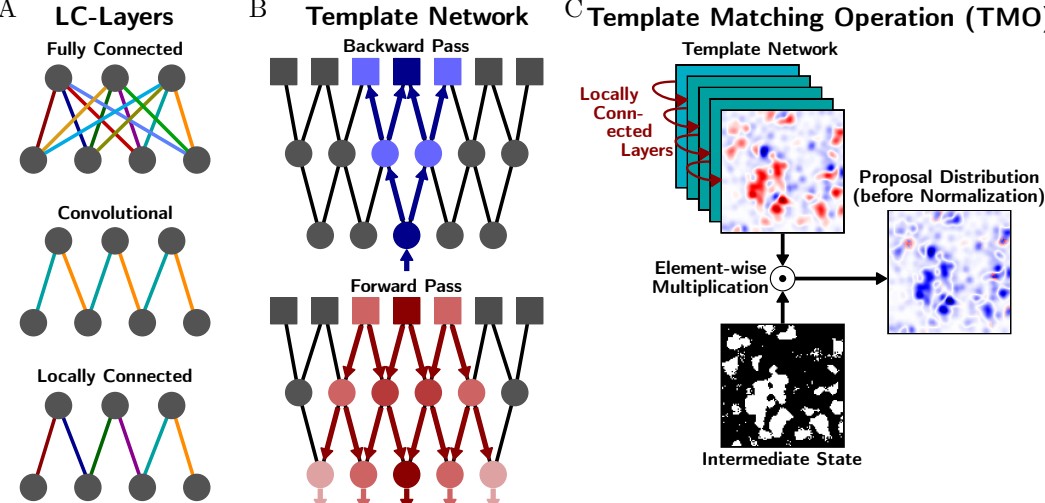

Figure 2: **A:** Locally connected networks are connected in the same way as convolutional layers but do not utilize weight sharing. **B:** Template networks first layer are learned parameters (squared boxes), followed by multiple 2D-locally connected layers (here shown in 1D). During the backward pass gradients are flowing from single positions to an increasing field of view from layer to layer. The next iteration's forward pass thus yields activations in the form of a Gaussian. **C:** The template matching operation (TMO) compares the learned template (network output) with the current state to calculate inversion probabilities by element-wise multiplication. Red: positive, blue: negative.

## 4 EXPERIMENTS

While our method is applicable for arbitrary functionalities (defined by corresponding fitness functions), we focus on analyzing properties of MeMe when searching for a 3-port wavelength demultiplexer (WDM; see Appx. A.4 for results for other optical functionalities). The WDM comprises one input and three output connections aiming to direct input signals of specific wavelengths to

different outputs (cf. Fig. 1A). We conduct experiments for $d_x = d_y = 100$, i.e., grids of $10,000$ elements resulting in $2^{10,000}$ possible states. As discussed above, we fix the learning rate $\eta = 1$ for all experiments and initialize the learned parameter layer to zero and locally connected layers weights to $1/k^2$. We set the connectivity of locally connected layers to $k = 2$ for all experiments and the network depth to $d = 10$ if not stated differently. Please see Appx. A.13 for details regarding the physical system as well as the FDFD-simulations and Appx. A.14 for exact definitions of fitness functions $\mathcal{F}$. Since we focus on algorithmic properties of MeMe in this publication, we utilize faster 2D-FDFD-simulations when evaluating the fitness functions. However, we validate the applicability of MeMe for 3D-simulations in Appx. A.5, where hyperparameters $T_S$ and $T_M$, which where optimized for 2D-systems can be used without further tuning. If not stated differently, we set $t_{max} = 10^6$ and use parallel exploration for MeMe and SA with multiple CPU-workers solving FDFD-simulations in parallel to evaluate inversion candidates (see Appx. A.10 for details on parallelization). Further experiments and visualizations offering additional insights to MeMe and its applications are given in the appendix.

## 4.1 OPTIMIZATION PROCESS

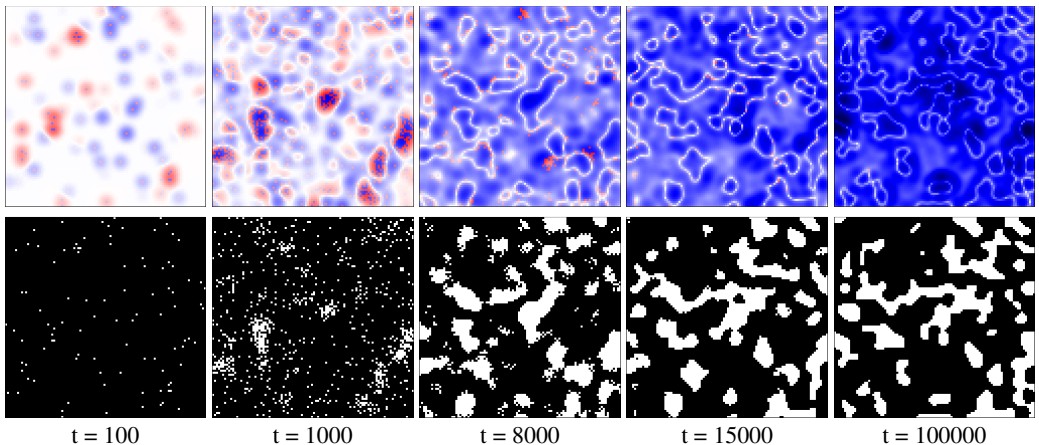

| t = 100 | t = 1000 | t = 8000 | t = 15000 | t = 100000 |

Figure 3: Optimization Process for $t_{max} = 10^5$, $T_S = 0.0001$ and $T_M = 0.1$. **Top**: TMO outputs $\boldsymbol{Q}_t$, i.e., proposal distribution before scaling and normalization. Blue: negative, red: positive, white: 0. **Bottom**: States $\boldsymbol{S}_t$ of stochastic process. Black: "1"/material, white: "-1"/air. Exploration followed by cluster formation and condensing to dense clusters.

To visualize the optimization process we depict the evolution of states $\boldsymbol{S}_t$ as well as the output of the template network after TMO (i.e. $\boldsymbol{Q}_t$) in Fig. 3. Starting from $\boldsymbol{S}_0^{ij} = 1$ and $\boldsymbol{Q}_0^{ij} = 0\ \forall i, j$ inversion proposals are sampled from $\boldsymbol{P}_t$ and are optionally reverted by the Metropolis criterion. Since the sampling distribution $\boldsymbol{P}_t$ is uniform in the first iteration, the optimization process is dominated by the Boltzmann distribution of fitness values $f_t$ in the beginning. Typically, most inversions are accepted during the early phase. Each action, reverted or not, propagates the corresponding rewards through the template network causing clusters with shapes resembling isotropic Gaussian functions in $\boldsymbol{Q}_t$ of the same sign as the reward. Accepted actions result in center pixels of opposite sign due to the sign flip in the TMO. By this, the template network identifies regions of high reward as well as detrimental regions. Subsequent actions will be sampled with higher probability from the proposed beneficial regions, resulting in first cluster formation ($t = 1000$). Next, clusters are filled progressively while larger structures are building. Due to the product loss (Eq. 2) the network output grows steadily (corresponding heatmaps getting darker), causing the proposal distribution to further deviate from a uniform distribution with every step. Since we do not regard the asymmetry of the proposal distribution when evaluating the Metropolis criterion as discussed above, this results in an increasing bias of the state distribution $\boldsymbol{S}_t$ from the Boltzmann distribution of fitness values $f_t$ towards the pattern learned by the template network, thus encouraging cluster formation. During the last phase of the optimization process cluster positions are mostly staying constant, while actions are mainly sampled from borders of clusters where subtle structures are learned and last freestanding pixels are incorporated into clusters. During this last phase, formation of sharp/edged structures

takes place, caused by decoupling of weights in locally connected layers. We further analyze this decoupling in Appx. A.2. Finally, the state $S_t$ converges against the exact learned template, i.e., the template network output (before TMO) $N_t$ thresholded at 0, for $T_M = \infty$, while finite $T_M$ result in a local maximum of $f_t$ where small differences from the template might still occur.

## 4.2 COMPARISON AGAINST SA AND REGULARIZED SA

Not only are isolated pixels not suited for actual optical chip fabrication, larger clustered regions are also reducing scattering and thus signal loss (Hughes et al., 2005) and are further easing fabrication. We chose to quantify these properties by defining the granularity $g(S_t)$ of a state as the normalized average number of pixels with dissimilar value in a 3x3 neighborhood (see Appx. A.1 for exact definition).

We conducted extensive experiments for combinations of the sampling temperature $T_S$ and the metropolis temperature $T_M$ and report results in Fig. 4A. For each temperature combination we perform optimizations with three random seeds and depict fitness $f$ and granularity $g$ for the state of the highest fitness per temperature combination. We seek to achieve high fitness $f$ while keeping the granularity $g$ low. As baseline results we compare against simulated annealing (SA) which equals setting $T_S = \infty$.

Furthermore, we show resulting states in Fig. 4 for the highest fitness result for SA (1), the highest fitness result for MeMe (2) and lowest granularity result (3). While SA reaches high fitness values for optimal $T_M = 0.0001$, resulting states (1) are not suited for actual fabrication due to many isolated pixels. On the other hand, setting $T_M = \infty$ yields states with dense clusters and without isolated pixels while fitness values are still comparably high. Combinations of finite $T_S$ and $T_M$ not only show cluster formation, but also reach fitness values which could not be reached by SA.

As an alternative approach, we compare against explicitly reducing the granularity by incorporating direct regularization into SA, i.e., distorting the energies/fitness values of the Boltzmann distribution of SA to $\tilde{f} = f - \lambda g$ with regularization factor $\lambda$. In Fig. 4B we show the full optimization trajectories for scanning $\lambda$ on (approximately) logarithmic scale and compare to MeMe's trajectory. Furthermore, we show (diagonal) contour lines for the regularized fitness $\tilde{f} = f - \lambda^{cont} g$ for $\lambda^{cont} = 0.1$. We chose $\lambda^{cont} = 0.1$ for visualization purposes only and report an alternative visualization for all $\lambda$ in Appx. Fig. A.2.

While explicit regularization significantly reduces the granularity $g$, high values of $\lambda$ result in a mode collapse. Starting from low $f$ and high $g$, small $\lambda$ prioritize maximization of $f$ while $g$ mainly decreases towards the end of each run. Higher regularization ($\lambda > 0.17$) effectively minimizes $g$, however, high fitness values are not achieved. The mode collapse can be observed best for $\lambda = 0.17$ where two runs reach high $f$ with high $g$, while one run reaches low $g$ with low $f$ despite the runs only differing in the random seed.

MeMe does not suffer from mode collapse but instead reaches both, high $f$ and low $g$. Visible when studying the countour lines, MeMe is even more efficient in minimizing $\tilde{f} = f - \lambda^{cont}$ than the corresponding runs for $\lambda^{cont} = 0.1$, despite MeMe not having any information about $g$. MeMe encourages cluster formation without counteracting fitness maximization, since the proposal distribution of MeMe is not only encouraging arbitrary cluster formation as regularizing $g$ does, but instead is itself learned to maximize $f$ and inherently tends to form clusters.

## 4.3 TEMPLATE NETWORK DEPTH

The template network architecture inherently controls size and shape of learned clusters. Next to the connectivity $k$, the network depth (number of layers) allows to control the cluster size. Furthermore, different initializations of the locally connected layers encourage different local patters (also see Appx. A.6). Additionally, convolutional layers could be used to learn translationally invariant patterns.

We depict optimized devices (highest fitness states) when varying the number of locally connected layers in Fig. 5. While the general trend, i.e., more layers result in larger cluster, can already be concluded qualitatively from the shown examples, we further quantify this observation. As shown in Fig. 5A, deep networks result in lower granularity but also cause fitness degradation, however, an optimal network depth and therefore resulting cluster size, can be seen for $d = 10$. Thus, encouraging cluster formation does not need to be contrary to fitness optimization (also see Appx. A.8). To measure the cluster size, we first calculate the autocorrelation function of the best fitness states for 10 optimization runs and plot the mean radial autocorrelation in Fig. 5B. We calculate the radial distance

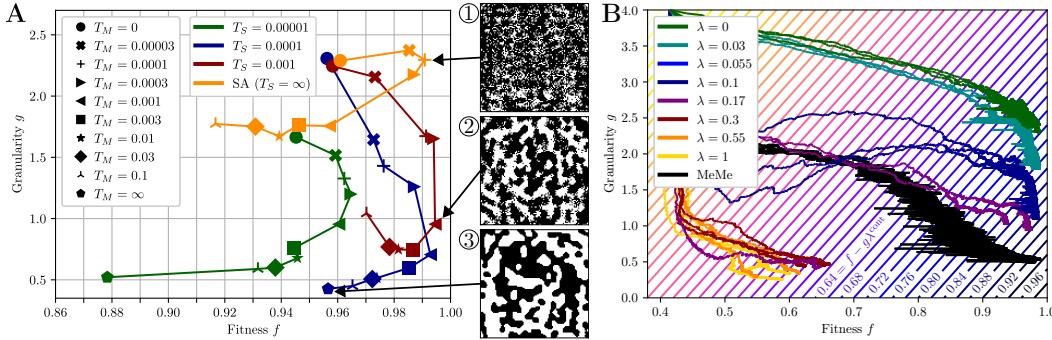

Figure 4: **A**: Hyperparameter scan of $T_S$ and $T_M$ including comparison against SA ($T_S = \infty$). Granularity $g$ (cf. Appx. A.1) and fitness $f$ for state with the highest fitness for each temperature combination. See Appx. Fig. A.13 for further results. MeMe reaches better fitness and lower granularity. **B**: Comparison against the Metropolis algorithm with additional granularity regularization, i.e., maximization of $\tilde{f} = f - \lambda g$. Diagonal lines mark contour of maximization objective for $\lambda^{cont} = 0.1$. $T_S = 0.0001$ and $T_M = 0.003$ (MeMe only). MeMe reaches lower values of $\tilde{f}$ compared to SA which explicitly minimizes $\tilde{f}$, despite having no information about $g$.

where the autocorrelation decayed to $50\%$ and depict the dependence on $d$ in Fig. 5C showing a sublinear cluster size growth with the network depth. Since the receptive field of each layer scales linearly with the network depth, linear growth would be an expected upper bound. When considering decoupling (cf. Appx. A.2) and finite size effects, the empirical observations thus aligns with our expectations. This confirms the tunability of the cluster size by varying the network depth.

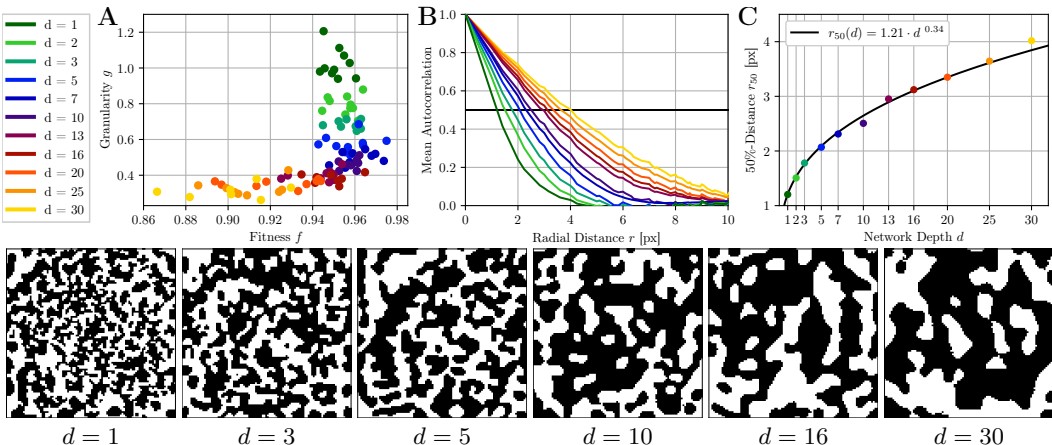

Figure 5: Dependence on network depth/number of locally connected layers $d$ for $T_S = 0.0001$ and $T_M = \infty$. **A**: $d$ controls fitness-granularity tradeoff. Best fitness values at $d = 10$. **B**: Radial autocorrelation function to measure cluster size. **C**: Cluster radius at $50\%$ autocorrelation vs. $d$. Sublinear dependence (exponent $0.34$ in least-squares fit). **Bottom Row**: Optimized highest fitness states.

## 5 CONCLUSION

We proposed MeMe, a novel algorithm for combinatorial optimization combining elements from Markov Chain Monte Carlo optimization and reinforcement learning utilizing novel template networks which encode optimal states without the need of input layers and are trained on single optimization runs. We demonstrate that MeMe finds clustered design patterns suitable for direct optical chip fabrication which can not be found by plain SA or regularized SA.

ACKNOWLEDGEMENTS

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

APPENDIX

A.1 DEFINITION OF GRANULARITY

We aim to quantify the number of isolated pixels but as well measure the formation of larger clustered regions. For this, we calculate the average number of grid cells with dissimilar cell state $\tilde{N}(\boldsymbol{S}_t)$ by convolving the state $\boldsymbol{S}_t$ with a 3x3 kernel $(K_{i,j})_{i,j\in\{-1,0,1\}}$ where $K_{0,0} = -1$ and $K_{i,j} = 1/8 \ \forall(i,j) \neq (0,0)$ and dividing by the grid size. Borders are padded with 1 to reflect the substrat-less surrounding of the chip. Directly using this definition for granularity calculations would not only disregard larger cluster formation, but also, if used as a regularizer, heavily favour the starting state impeding the Metropolis algorithm for large regularization factors $\lambda$. Thus, we normalize this metric by subtracting the starting state value (which is not 0 due to border cells) and normalize by the total ratio of cells with dissimilar state:

$$g(\boldsymbol{S}_t) = \frac{\tilde{N}(\boldsymbol{S}_t) - \tilde{N}(\boldsymbol{S}_0)}{1 - |\sum\limits_{i,j}^{d_x,d_y} \boldsymbol{S}_t^{i,j}|/(d_x d_y)} \tag{A.1}$$

A.2 DECOUPLING

As a measure of decoupling, i.e., the deviations of the stacked locally connected layers from a Gaussian filtering, we calculate the standard deviation of weights per spatial position. In analogy to a convolution layer, this is calculating the standard deviation per filter/kernels (which are not shared spatially in locally connected layers). However, there will also be more noise and thus higher standard deviations for filters/kernels which simply changed more compared to their initial value, thus, we additionally divide by the absolute difference from the initialization value and average over all layers. With weight $W_{ijnm}^{l,t}$ at layer $l$, iteration $t$, spatial positions $i, j$ and filter/kernel indices $n, m$ this is

$$D_{ij}^t = \frac{1}{L} \sum_{l=1}^{L} \frac{\sqrt{\frac{1}{NM} \sum\limits_{n,m} \left(\left(\frac{1}{NM} \sum\limits_{n,m} W_{ijnm}^{l,t}\right) - W_{ijnm}^{l,t}\right)^2}}{\sum\limits_{n,m} |W_{ijnm}^{l,t} - W_{ijnm}^{l,0}| + \epsilon}, \tag{A.2}$$

where $\epsilon = 10^{-6}$ is a constant for numerical stability. We depict the decoupling factor for the final iteration in Fig. A.1. The decoupling reaches highest values at the edges of the device, supporting that decoupling allows the template network to learn finegrained structures.

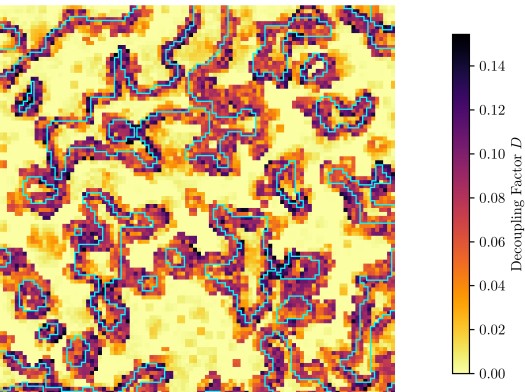

Figure A.1: Decoupling Factor (Eq. A.2) overlayed with device edges. $t_{max} = 10^6$, $T_S = 0.0001$ and $T_M = \infty$.

### A.3 MeMe vs. Regularized SA

We show that MeMe reaches higher regularized fitness values $\tilde{f} = f - \lambda g$ for all $\lambda$ despite MeMe not having any information about $g$, which we showed for $\lambda^{cont} = 0.1$ in Fig. 4B. Instead of drawing the full contour of $\tilde{f}$ as done in the main text, we only draw the contour line for the highest value of $\tilde{f}$ that could be reached with regularized SA, thus the end points of optimizations with regularized SA are intercepting with these lines (see Fig. A.2).

MeMe reaches points below all of these lines (i.e. bottom right corner) and hence surpasses regularized SA for any $\lambda$.

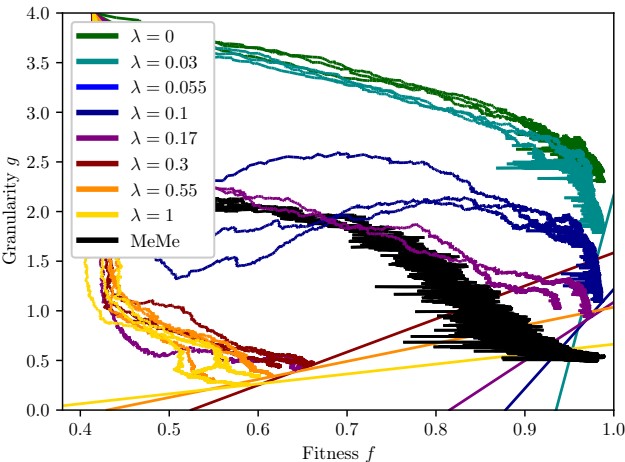

Figure A.2: All $\lambda$ results for Figure 4B.

### A.4 Further devices compared against regularized SA

We compare MeMe against granularity regularized SA for three photonic functionalities and report results in Fig. A.3. Namely, we compare results for 90-10 power splitters (PS), mode demultiplexers (MDM), and the wavelength demultiplexer (WDM) discussed in the main text (see Appx. A.14 for corresponding fitness functions). We set $T_S = 0.0001$ and $t_{max} = 10^6$. For regularized SA we scan the regularization parameter $\lambda$ and show results with highest fitness for the highest $\lambda$ before mode collapse (cf. Fig. 4), i.e. $\lambda_{\text{WDM}} = 0.17$, $\lambda_{\text{WDM}} = 0.17$ and $\lambda_{\text{PS}} = 0.3$.

While SA reaches comparable results to MeMe, many isolated pixels and small clusters occur, making these devices unsuitable for actual chip fabrication.

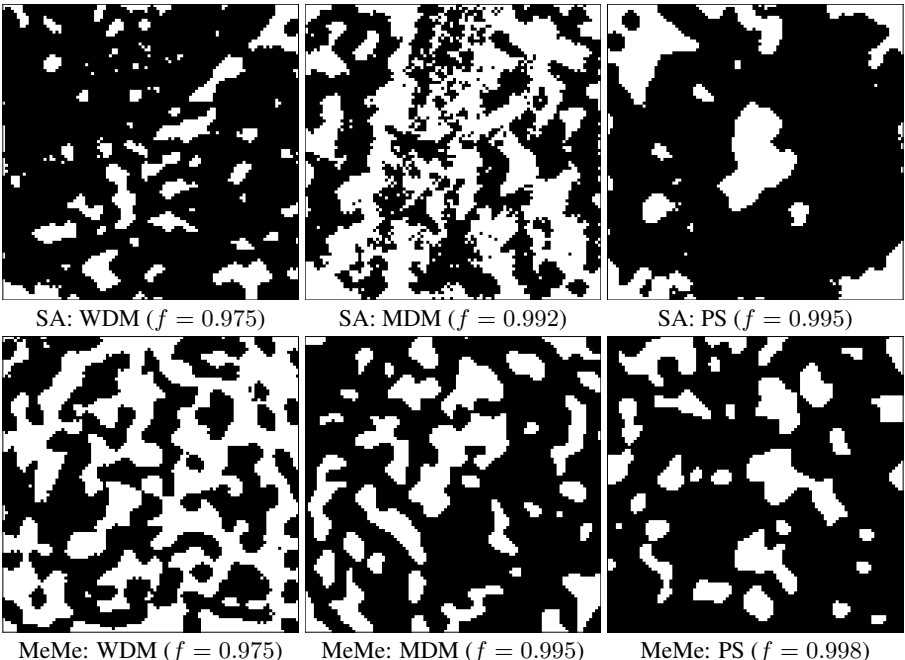

SA: WDM ($f = 0.975$)  SA: MDM ($f = 0.992$)  SA: PS ($f = 0.995$)

MeMe: WDM ($f = 0.975$)  MeMe: MDM ($f = 0.995$)  MeMe: PS ($f = 0.998$)

Figure A.3: Best devices found for MeMe and SA with granularity regularization. MeMe reaches high fitness values for all devices while yielding densely clustered structures.

## A.5 3D FDFD SIMULATIONS

We utilized 2D-FDFD-simulations in the main text, since we focused on algorithmic properties of MeMe. While these simulations yield less accurate predictions of electromagnet fields, they require less computational resources and thus allowed us to perform the shown extensive empirical analysis of MeMe. However, we report results for a 90-10 power splitter (PS) evaluated with 3D-FDFD-simulations with $d_x = d_y = 80$, $T_S = 0.0001$, $T_M = 0.01$ and $t_{max} = 10^5$. While these simulations are computationally more costly, they yield high accuracy results directly transferable to experimental measurements.

We find an optimized state yielding normalized powers of $p_1 = 0.093$ and $p_2 = 0.845$ (see Fig. A.4). Furthermore, we show the optimization trajectory in Fig. A.5.

By this we confirm the applicability of MeMe to 3D-FDFD-simulations. We are looking forward to extending these promising results to other optical functionalities and validate our findings with experimental measurements in future work.

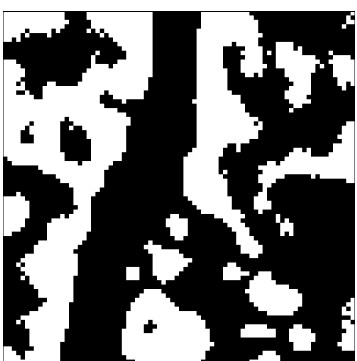

Figure A.4: Optimized state for PS with 3D-FDFD-simulations.

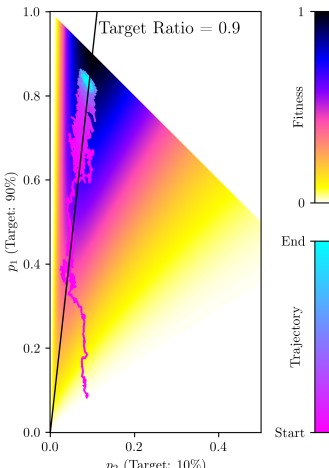

Figure A.5: 90-10 PS utilizing 3D-FDFD-simulations. Fitness function $\mathcal{F}(\boldsymbol{S})$ in dependence on corresponding powers $p_1(\boldsymbol{S}_t)$ and $p_2(\boldsymbol{S}_t)$ overlayed with optimization trajectory.

## A.6 NETWORK INITILIZATION PRIORS

The initialization of the first fully learned layer allows to set prior distributions. For power splitters (PS) a straightforward prior is to directly connect input and output waveguides. MeMe yields final states close to the prior distribution (cf. Fig. A.6). Setting priors can reduce the optimization runtime significantly as shown in Fig. A.7.

However, we could not find equally high fitness values as without setting a prior, probably caused by steering into a local minimum where more complex patterns (cf. Fig. A.4) could not be reached anymore. Furthermore, finding a reasonable prior might be challenging for other devices such as wavelength demultiplexers (WDM) and mode demultiplexers (MDM).

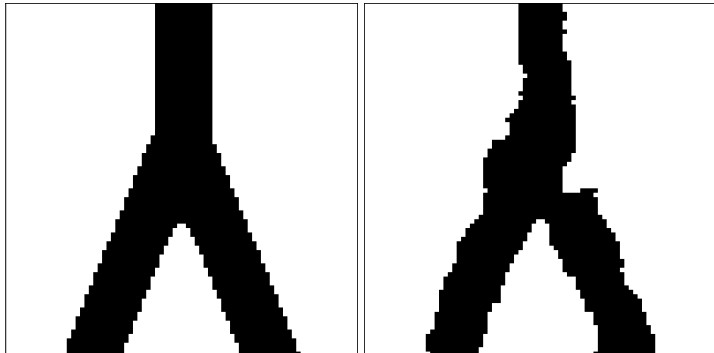

Figure A.6: **Left**: Prior state directly connecting input and output waveguides. **Right**: Optimized state. 90-10-power splitter utilizing 3D-FDFD-simulations.

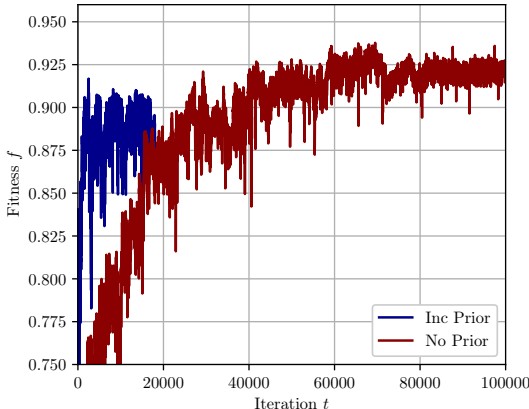

Figure A.7: MeMe optimization with and without setting prior as in Fig.A.6. PS utilizing 3D-FDFD-simulations. Prior causes the best device to be found after less than 2000 iterations, however fitness values are lower.

## A.7 LARGER PIXEL SIZE

Alternative to encouraging cluster formation of small pixels, the pixel size could be simply increased. However, this results in dramatically restricting the search space. Optimizing with SA thus does not yield comparable fitness values to MeMe results (See Fig. A.8)

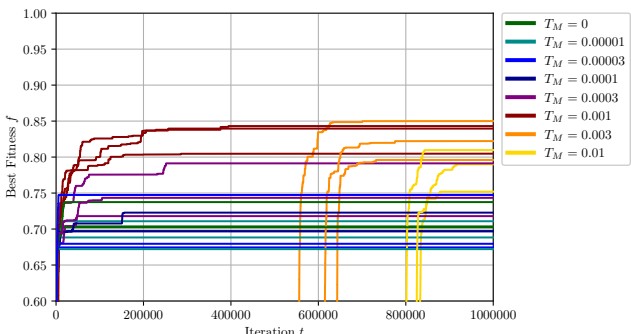

Figure A.8: Accumulated maximal fitness for SA with 4x increased pixel size. Three random seeds each. $t_{max} = 10^6$.

## A.8 SPATIAL REWARD PATTERNS

An unbiased Metropolis algorithm samples states according to the Boltzmann distribution of fitness values (energies). This is equivalent to sampling transitions from a Boltzmann distribution of the fitness differences ($r_t = f_{t+1} - f_t$) which we use as rewards for the RL agent. However, fully calculating this transition probability requires to evaluate the reward for every possible action which results in $d_x \cdot d_y = 10000$ FDFD-simulations. While the acceptance criterion of the Metropolis algorithm allows sampling from this distribution without explicitly calculating it, we here calculate the rewards for all possible actions in one state. Since rewards are fitness differences and thus are symmetric, we depict $\boldsymbol{R}_t \odot \boldsymbol{S}_t$ to visualize the rewards independent of the current pixels state.
In Fig. A.9 we show rewards for corresponding pixel inversions if starting from an all-material state (left). Even though without clear symmetry, a clustered pattern occurs. However, when finding a local maximum via SA (i.e. all rewards $\boldsymbol{R}_t$ are negative), not only state pixels are mostly isolated (mid) but also reward distributions are not showing cluster formation anymore (right). The isolated pixel inversions of SA seem to break up the clustered patterns in the rewards (and thus in the Boltzmann transition distribution) and thus cause final states to consist of many isolated pixels too.

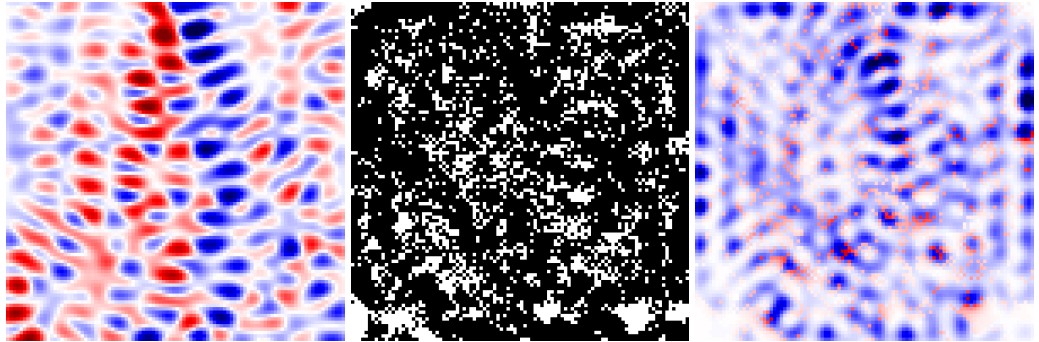

Figure A.9: **Left**: $\boldsymbol{R}_t \odot \boldsymbol{S}_t$ at $t = 0$, i.e., for all-material state. **Mid**: Local fitness maximum obtained via SA. State $\boldsymbol{S}_t$ at $t = t_{max}$. **Right**: $\boldsymbol{R}_t \odot \boldsymbol{S}_t$ at $t = t_{max}$. Even though initial distribution was clustered, isolated pixels of SA states cause breaking of clustered structure and thus no clusters occur in final state.

When optimizing with MeMe, the template network learns a template targeted at predicting the rewards (and thus the Boltzmann transition distribution). We depict the resulting reward pattern next to the template network output in Fig. A.10. The template network learns to predict the full reward pattern with high accuracy while maintaining a clustered pattern. Furthermore, the rewards pattern itself does not show isolated pixels as for the SA-optimized device.

The states of the resulting stochastic process of MeMe are sampled from a distribution combining the learned template (mid) and the actual reward distribution (right) controlled by $T_S$ and $T_M$. The learned template encourages cluster formation, however, since it itself approximates the reward, high fitness values can be achieved.

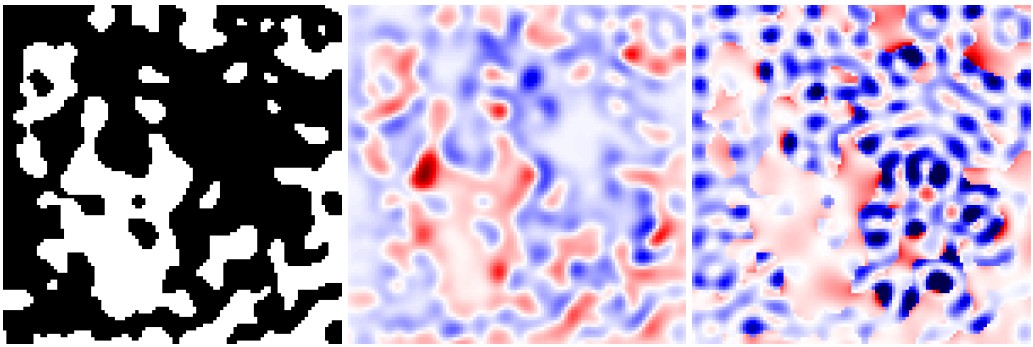

Figure A.10: **Left**: State $\boldsymbol{S}_t$. **Mid**: Network output $\boldsymbol{N}_t$. **Right**: Reward for each pixel inversion pixel-wise multiplied with state ($\boldsymbol{R}_t \odot \boldsymbol{S}_t$; colormap thresholded for better visualization). All at $t = t_{max}$ with $t_{max} = 10^5$, $T_S = 0.0001$ and $T_M = 0.1$.

## A.9 UNBIASED SAMPLING

To construct a stochastic process that generates unbiased samples from the underlying Boltzmann distribution of fitness values, the proposal distribution asymmetry has to be considered when evaluating the Metropolis criterion (see Alg. A.1). With a slight abuse of notation this is scaling the rejection probability by $\boldsymbol{P}_t(a_t^{-1}|\tilde{\boldsymbol{S}}_t)/\boldsymbol{P}_t(a_t|\boldsymbol{S}_t)$, where $\boldsymbol{P}_t(a_t|\boldsymbol{S}_t)$ notates the probability for sampling action $a_t$ in state $\boldsymbol{S}_t$ and where $a_t$ describes the transition $\boldsymbol{S}_t \rightarrow \tilde{\boldsymbol{S}}_t$. The inverted action $a_t^{-1}$ describes $\tilde{\boldsymbol{S}}_t \rightarrow \boldsymbol{S}_t$, however, $a_t^{-1} = a_t$ since actions are pixel inversions.

Resulting granularities and fitness values in analogy to Fig. 4A are shown in Fig. A.11. Compared to SA, fitness values $f$ could be increased marginally, however, the granularity is not reduced, i.e., no increased cluster formation occurs.

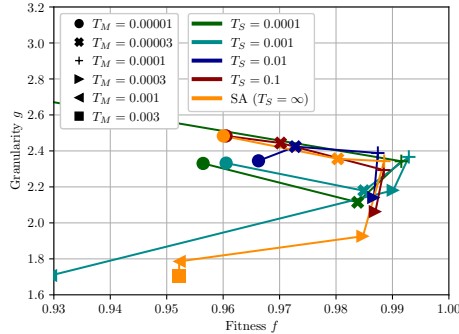

Figure A.11: Unbiased sampling from Boltzmann distribution. No increased cluster formation compared to SA.

---

**Algorithm A.1:** MeMe with unbiased sampling

---

**Input:** Design area dimensions $d_x$ and $d_y$, sampling temperature $T_S$, metropolis base
temperature $T_M^0$, learning rate $\eta$, iterations $t_{max}$, fitness function $\mathcal{F}$, loss function $\mathcal{L}$
**Initialize:** State $\boldsymbol{S}_0 \in \{-1, 1\}^{d_x \times d_y}$ here: $\boldsymbol{S}_0^{i,j} = 1 \, \forall i, j$; network parameters $\Theta_0$
**for** $t \leftarrow 0$ **to** $t_{max}$ **do**
    $\boldsymbol{N}_t = \mathcal{N}(\Theta_t)$                           `// evaluate template network`
    $\boldsymbol{Q}_t = -\boldsymbol{N}_t \odot \boldsymbol{S}_t$                    `// template matching operation (TMO)`
    $\boldsymbol{P}_t^{ij} = \log(1 + \exp(\boldsymbol{Q}_t^{ij}/T_S))/\sum_{i,j} \log(1 + \exp(\boldsymbol{Q}_t^{ij}/T_S))$     `// normalization`
    $a_t = (a_t^x, a_t^y) \sim \boldsymbol{P}_t$                   `// sample action (pixel indices)`
    $\tilde{\boldsymbol{S}}_t = \boldsymbol{S}_t$                                    `// copy state`
    $\tilde{\boldsymbol{S}}_t^{a_t^x, a_t^y} = -1 \cdot \tilde{\boldsymbol{S}}_t^{a_t^x, a_t^y}$                   `// invert pixel`
    $\tilde{f}_t \leftarrow \mathcal{F}(\tilde{\boldsymbol{S}}_t)$       `// evaluate fitness function (FDFD simulation)`
    $r_t = \tilde{f}_t - f_t$                        `// calculate reward`
    $\Theta_{t+1} = \Theta_{t+1} - \eta \partial \mathcal{L}(\boldsymbol{Q}_t, r_t, a_t))/\partial \Theta_t$       `// gradient descent`
    $T_M^t = T_M^0(1 + \cos(\pi t/t_{max}))/2$         `// simulated annealing`
    **if** $\exp(r_t/T_M^t) \cdot \boldsymbol{P}_t(a_t^{-1}|\tilde{\boldsymbol{S}}_t)/\boldsymbol{P}_t(a_t|\boldsymbol{S}_t) < X \sim \mathcal{U}(0,1)$ **then**     `// metropolis`
     `criterion`
        $\boldsymbol{S}_{t+1} = \boldsymbol{S}_t$                       `// revert action`
        $f_{t+1} = f_t$
    **else**
        $\boldsymbol{S}_{t+1} = \tilde{\boldsymbol{S}}_t$                     `// accept action`
        $f_{t+1} = \tilde{f}_t$
    **end**
**end**
**return** $S_{t_{max}}$

---

## A.10 PARALLEL EXPLORATION

We implement SA and MeMe with parallel exploration, i.e., multiple workers are solving computationally costly FDFD-simulations in parallel, however, actions are still applied sequentially. The corresponding adapted algorithms for SA and MeMe are shown in Alg. A.2 and Alg. A.3. As a consequence we calculate the reward for each pixel as the difference to the fitness at the beginning of the last batch ($r_t = \tilde{f}_t - f_j$) instead of the last iteration ($r_t = \tilde{f}_t - f_{t+1}$), causing minor divergences. However, states and fitness values are always matched correctly. We set $N = 24$ for all results in the main text and $N = 1$ for 3D-FDFD-simulation results in Appx. A.5.
We empirically study the effect of the number of workers $N$ exploring in parallel in Fig. A.12. While fitness values $f$ are not effected, larger $N$ tend to lower granularity devices, however, 2-$\sigma$ confidence ellipses overlap for all $N$.

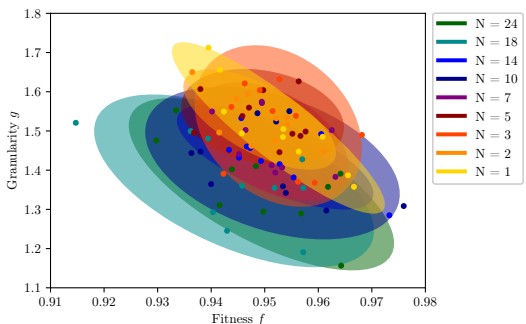

Figure A.12: Parallel exploration with $N$ workers. 10 random seeds per $N$. 2-$\sigma$ confidence ellipses. $T_M = 0.0001, T_S = 0.00001, t_{max} = 10^5$.

---

**Algorithm A.2:** Simulated Annealing with parallel exploration

---

**Input:** Design area dimensions $d_x$ and $d_y$, metropolis base temperature $T_M^0$, iterations $t_{max}$,
      number of workers $N$

**Initialize:** State $\boldsymbol{S}_0 \in \{-1, 1\}^{d_x \times d_y}$ here: $\boldsymbol{S}_0^{i,j} = 1 \; \forall i, j$

**for** $j \leftarrow 0$ **to** $t_{max}/N$ **do**
    $t = j \cdot N$
    **for** $n \leftarrow 0$ **to** $N$ **do**             // execute in parellel
        $t \leftarrow t + 1$
        $a_t = (a_t^x, a_t^y) \sim \mathcal{U}$       // sample action uniformly
        $\tilde{\boldsymbol{S}}_t = \boldsymbol{S}_j$         // copy state
        $\tilde{\boldsymbol{S}}_t^{a_t^x, a_t^y} = -1 \cdot \tilde{\boldsymbol{S}}_t^{a_t^x, a_t^y}$         // invert pixel
        $\tilde{f}_t \leftarrow \mathcal{F}(\tilde{\boldsymbol{S}}_t)$         // evaluate fitness
    **end**
    $f_j \leftarrow \mathcal{F}(\boldsymbol{S}_j)$       // evaluate in parallel to workers
    *join_workers()*
    $t = j \cdot N$       // reset t for next loop
    **for** $n \leftarrow 0$ **to** $N$ **do**       // execute sequentially
        $t \leftarrow t + 1$
        $\boldsymbol{S}_{t+1} = \boldsymbol{S}_t$         // copy state
        $r_t = \tilde{f}_t - f_j$         // calculate reward
        **if** $\exp(r_t/T_M^t) < X \sim \mathcal{U}(0,1)$ **then**     // metropolis criterion
            **pass**         // no action
        **else**
            $\boldsymbol{S}_{t+1}^{a_t^x, a_t^y} = -1 \cdot \boldsymbol{S}_{t+1}^{a_t^x, a_t^y}$         // apply action
        **end**
    **end**
**end**
**return** $S_{t_{max}}$

---

---

**Algorithm A.3:** MeMe with parallel exploration

---

**Input:** Design area dimensions $d_x$ and $d_y$, sampling temperature $T_S$, metropolis base
temperature $T_M^0$, learning rate $\eta$, iterations $t_{max}$, number of workers $N$
**Initialize:** State $\boldsymbol{S}_0 \in \{-1, 1\}^{d_x \times d_y}$ here: $\boldsymbol{S}_0^{i,j} = 1 \,\forall i, j$
**for** $j \leftarrow 0$ **to** $t_{max}/N$ **do**
   $t = j \cdot N$
   **for** $n \leftarrow 0$ **to** $N$ **do**                         `// execute in parellel`
      $t \leftarrow t + 1$
      $\boldsymbol{N}_t = \mathcal{N}(\Theta_t)$               `// evaluate template network`
      $\boldsymbol{Q}_t = -\boldsymbol{N}_t \odot \boldsymbol{S}_t$       `// template matching operation (TMO)`
      $\boldsymbol{P}_t^{ij} = \log(1 + \exp(\boldsymbol{Q}_t^{ij}/T_S))/\sum_{i,j}\log(1 + \exp(\boldsymbol{Q}_t^{ij}/T_S))$   `// normalization`
      $a_t = (a_t^x, a_t^y) \sim \boldsymbol{P}_t$       `// sample action (pixel indices)`
      $\tilde{\boldsymbol{S}}_t = \boldsymbol{S}_j$                       `// copy state`
      $\tilde{\boldsymbol{S}}_t^{a_t^x, a_t^y} = -1 \cdot \tilde{\boldsymbol{S}}_t^{a_t^x, a_t^y}$       `// invert pixel`
      $\tilde{f}_t \leftarrow \mathcal{F}(\tilde{\boldsymbol{S}}_t)$               `// evaluate fitness`
   **end**
   $f_j \leftarrow \mathcal{F}(\boldsymbol{S}_j)$            `// evaluate in parallel to workers`
   *join_workers()*
   $t = j \cdot N$                       `// reset t for next loop`
   **for** $n \leftarrow 0$ **to** $N$ **do**          `// execute sequentially`
      $t \leftarrow t + 1$
      $\boldsymbol{S}_{t+1} = \boldsymbol{S}_t$                 `// copy state`
      $r_t = \tilde{f}_t - f_j$             `// calculate reward`
      $\Theta_{t+1} = \Theta_{t+1} - \eta \partial \mathcal{L}(\boldsymbol{Q}_t, r_t, a_t))/\partial \Theta_t$     `// gradient descent`
      **if** $\exp(r_t/T_M^t) < X \sim \mathcal{U}(0,1)$ **then**    `// metropolis criterion`
         **pass**                         `// no action`
      **else**
         $\boldsymbol{S}_{t+1}^{a_t^x, a_t^y} = -1 \cdot \boldsymbol{S}_{t+1}^{a_t^x, a_t^y}$       `// apply action`
      **end**
   **end**
**end**
**return** $S_{t_{max}}$

---

## A.11 ALL SEEDS RESULTS FOR TEMPERATURE SCAN

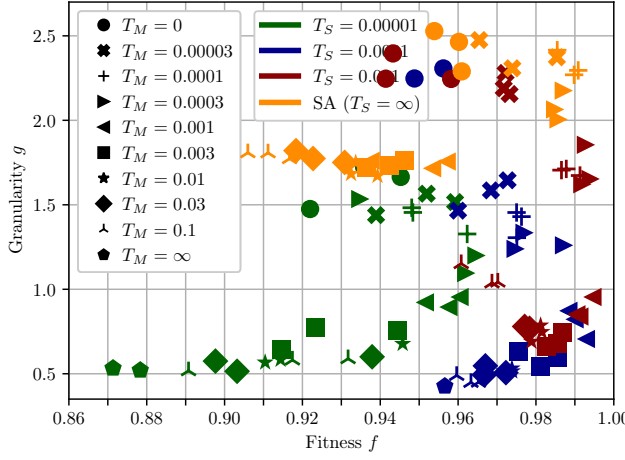

Figure A.13: All seed results for Fig. 4A.

## A.12 SIMULATED ANNEALING (SA) ALGORITHM

---

**Algorithm A.4:** Simulated Annealing

---

**Input:** Design area dimensions $d_x$ and $d_y$, metropolis base temperature $T_M^0$, iterations $t_{max}$
**Initialize:** State $\boldsymbol{S}_0 \in \{-1, 1\}^{d_x \times d_y}$ here: $\boldsymbol{S}_0^{i,j} = 1 \; \forall i, j$
**for** $t \leftarrow 0$ **to** $t_{max}$ **do**
     $a_t = (a_t^x, a_t^y) \sim \mathcal{U}$                             // sample action uniformly
     $\tilde{\boldsymbol{S}}_t = \boldsymbol{S}_t$                                     // copy state
     $\tilde{\boldsymbol{S}}_t^{a_t^x, a_t^y} = -1 \cdot \tilde{\boldsymbol{S}}_t^{a_t^x, a_t^y}$                  // invert pixel
     $\tilde{f}_t \leftarrow \mathcal{F}(\tilde{\boldsymbol{S}}_t)$       // evaluate fitness function (FDFD simulation)
     $r_t = \tilde{f}_t - f_t$                            // calculate reward
     $T_M^t = T_M^0 (1 + \cos(\pi t / t_{max}))/2$          // simulated annealing
     **if** $\exp(r_t / T_M^t) < X \sim \mathcal{U}(0, 1)$ **then**     // metropolis criterion
         $\boldsymbol{S}_{t+1} = \boldsymbol{S}_t$                          // revert action
         $f_{t+1} = f_t$
     **else**
         $\boldsymbol{S}_{t+1} = \tilde{\boldsymbol{S}}_t$                        // accept action
         $f_{t+1} = \tilde{f}_t$
     **end**
**end**
**return** $S_{t_{max}}$

---

## A.13 PHYSICAL DETAILS AND FDFD-SIMULATIONS

The electromagnetic simulations in this work have been conducted using the finite-difference frequency-domain (FDFD) method. Our implementation of method solves the time-harmonic Maxwell equation shown in Eq. A.3 for the electric field vector.

$$\nabla \times \mu_0^{-1} \nabla \times \vec{E} - \omega^2 \epsilon \vec{E} = -i\omega \vec{J}, \tag{A.3}$$

where $\mu_0$ and $\epsilon$ are the magnetic permeability and electric permittivity of the material at the respective grid cell, $\omega$ is the angular frequency and $\vec{J}$ is the current distribution of the source. Using space-discretized operators, Eq. A.3 can be transformed into a system of coupled equations which is expressed as

$$\mathbf{A} \cdot \vec{x} - \vec{b} = \vec{0}. \tag{A.4}$$

Here, the matrix $\mathbf{A}$ represents the physical properties of the system, $\vec{b}$ is the excitation of the system, and $\vec{x}$ is the solution vector containing the components of the electric field distribution in the simulation cell.

The simulation cell, which includes the design area and the surrounding waveguide geometry, is numerically respresented using a Yee-Grid discretization scheme (Yee, 1966). The grid-cells have uniform dimensions with a cell size of $50\,\text{nm}$. As a boundary condition, we employ a perfectly matched layer (PML) of sufficient thickness ($> 0.5\,\lambda$) surrounding the simulation cell.

The current sources of the conducted simulations are transverse-electric (TE) modes. All devices have been simulated for a fundamental TE-mode current source in the input waveguide. For the mode-demultiplexer (MDM), a second waveguide mode source has been considered representing the $TE_{10}$-mode. The wavelenght of the FDFD-simulations is $\lambda = 1550\,\text{nm}$. The second and third wavelength considered for the wavelength-demultiplexer (WDM) are $\lambda = 1540\,\text{nm}$ and $\lambda = 1530\,\text{nm}$, respectively.

The photonic platform employed is silicon-on-insulator. Tab. A.1 shows the refractive index values (Li, 1980).

The 2D-simulations have been conducted using an effective refractive index approximation to reduce the impact of the high difference between the effective wavelengths in 2D and 3D simulations, respectively.

Table A.1: Refractive indices of silicon at the cosidered wavelengths.

| Wavelength ($\lambda$) | Refractive Index ($n_{\text{Si}}$) |
|---|---|
| 1550 nm | 3.4757 |
| 1540 nm | 3.4765 |
| 1530 nm | 3.4774 |

## A.14  FITNESS FUNCTIONS

The devices shown in this work have been optimized using the following fitness functions. $\eta_{i \to j}$ denotes the fraction of the total power inserted into the system through input-mode $i$ being coupled into mode $j$ at the target output-port. $\text{CT}_i$ is the crosstalk of the other wavelengths or modes at output port $i$.

### A.14.1  MODE DEMULTIPLEXER AND WAVELENGTH DEMULTIPLEXER

For the mode demultiplexer (MDM), we consider fundamental, second order and third order transverse electric waveguide modes at $\lambda = 1550\,\text{nm}$ as inputs to the system. In case of the wavelength demultiplexer (WDM), the input modes are all fundamental, but the wavelengths are shifted such that $\lambda_0 = 1530\,\text{nm}$, $\lambda_1 = 1540\,\text{nm}$ and $\lambda_2 = 1550\,\text{nm}$. The fitness function calculates the harmonic mean of the fitness functions at each individual output port, which are a sums of the desired coupling efficiencies subtracted by the calculated crosstalk resulting from the other inputs.

$$\mathcal{F}_{\text{WDM, MDM}}(\boldsymbol{S}) = 3 \left( \frac{2}{\eta_{0 \to 0} - \mathbf{CT}_0 + 1} + \frac{2}{\eta_{1 \to 1} - \mathbf{CT}_1 + 1} + \frac{2}{\eta_{2 \to 2} - \mathbf{CT}_2 + 1} \right)^{-1} \quad \text{(A.5)}$$

### A.14.2  POWER SPLITTER

Since there is only one input mode for the power splitter (PS), we denote the corresponding output powers $\eta_0$ and $\eta_1$. With target ratio $\tilde{r} = 0.9$ for a 90-10-PS we set

$$\mathcal{F}_{\text{PS}}(\boldsymbol{S}) = \exp\left( -\left( \frac{\tilde{r}}{\eta_0/\eta_1} + \frac{\eta_0/\eta_1}{\tilde{r}} \right)/2 - 1 \right) \cdot (\eta_0 + \eta_1). \quad \text{(A.6)}$$

## A.15  EFFECT OF MARKOV PROPERTY VIOLATION

Violating the Markov property by dropping the correction term for the asymmetry of the proposal distribution biases the generated samples of the random process and thus the converging state towards the proposal distribution. We are analyzing this effect in more detail here and analytically derive the exact distribution from which samples are generated.

In the following we will drop the temporal index $t$ for better readability. The analysis can be done analogously for softplus and softmax normalization of TMO outputs $\boldsymbol{Q}$, however, for simplicity we will focus on softmax normalization here, which also approximates softplus normalization since $\log(1 + x) \approx x$ for $x \approx 0$. Please also see Sec. A.9 for unbiased sampling.

With a slight abuse of notation the correction would be to scale the rejection probability by $\boldsymbol{P}(a^{-1}|\tilde{\boldsymbol{S}})/\boldsymbol{P}(a|\boldsymbol{S})$, where $\boldsymbol{P}(a|\boldsymbol{S})$ notates the probability for sampling action $a$ in state $\boldsymbol{S}$ and where $a$ describes the transition $\boldsymbol{S} \to \tilde{\boldsymbol{S}}$. The inverted action $a^{-1}$ describes $\tilde{\boldsymbol{S}} \to \boldsymbol{S}$, however, $a^{-1} = a$ since actions are pixel inversions (i.e. tuples of indices, $a \in \{(i,j)|0 \le i < d_x; 0 \le j < d_y; i,j \in \mathbb{N}\}$).

Dropping this correction term equals dividing by it, i.e. the unbiased (unnormalized) Boltzmann distribution $\exp(r_t/T_M^t)$ is scaled by $\boldsymbol{P}(a|\boldsymbol{S})/\boldsymbol{P}(a^{-1}|\tilde{\boldsymbol{S}})$. Reintroducing the index notation for the action as used in the main text yields

$$\boldsymbol{B}_{ij} := \boldsymbol{P}_{ij}(\boldsymbol{S})/\boldsymbol{P}_{ij}(\tilde{\boldsymbol{S}}), \quad \text{(A.7)}$$

which equals the (unnormalized) 2D distribution by which MeMe biases the stochastic process towards cluster formation. Given the softmax normalization and that $\tilde{\boldsymbol{S}}$ only differs from $\boldsymbol{S}$ by

flipping the sign of $\boldsymbol{Q}_{ij}$ it holds that

$$\boldsymbol{P}_{ij}(\boldsymbol{S}) = \frac{\exp(\boldsymbol{Q}_{ij}/T_S)}{\sum\limits_{n,m} \exp(\boldsymbol{Q}_{nm}/T_S)} \tag{A.8}$$

$$\boldsymbol{P}_{ij}(\tilde{\boldsymbol{S}}) = \frac{\exp(-\boldsymbol{Q}_{ij}/T_S)}{\exp(-\boldsymbol{Q}_{ij}/T_S) + \sum\limits_{n,m \neq i,j} \exp(\boldsymbol{Q}_{nm}/T_S)}. \tag{A.9}$$

Since $\exp(|\boldsymbol{Q}_{ij}/T_S|) \ll \sum\limits_{n,m} \exp(\boldsymbol{Q}_{nm}/T_S)$ it follows that

$$\boldsymbol{B}_{ij} = \boldsymbol{P}_{ij}(\boldsymbol{S})/\boldsymbol{P}_{ij}(\tilde{\boldsymbol{S}}) \approx \exp(\boldsymbol{Q}_{ij}/T_S)/\exp(-\boldsymbol{Q}_{ij}/T_S) = \exp(2\boldsymbol{Q}_{ij}/T_S). \tag{A.10}$$

Thus, the resulting stochastic process again fulfills the Markov property for the product of the Boltzmann distribution of fitness values and the bias $\boldsymbol{B}$ which monotonically depends on the TMO output $\boldsymbol{Q}$ (learned template). Furthermore we confirm that $T_S$ controls the induced bias.

## A.16 NOTATION SUMMARY

| | |
|---|---|
| state space | $\mathcal{S} = \{-1, 1\}^{d_x \times d_y}$ |
| state | $\boldsymbol{S}_t \in \mathcal{S}$ |
| action space | $\mathcal{A} = \{(i,j) \mid 0 \leq i < d_x; 0 \leq j < d_y; i,j \in \mathbb{N}\}$ |
| action | $a_t \in \mathcal{A}$ |
| fitness | $f_t \leftarrow \mathcal{F}(\boldsymbol{S}_t)$ |
| reward | $r_t = \tilde{f}_t - f_t$ |
| template network output | $\boldsymbol{N}_t = \mathcal{N}(\Theta_t)$ |
| template matching operation (TMO) | $\boldsymbol{Q}_t = -\boldsymbol{N}_t \odot \boldsymbol{S}_t$ |
| normalized proposal distribution (policy) | $\boldsymbol{P}_t$ |

