# OpenReview forum: "Combinatorial Optimization via Memory Metropolis: Template Networks for Proposal Distributions in Simulated Annealing applied to Nanophotonic Inverse Design"
_ICLR.cc/2024/Conference — Submitted to ICLR 2024_

### Official Review · Reviewer_9yfn · 2023-10-31

**Soundness:** 2 fair
**Presentation:** 2 fair
**Contribution:** 2 fair
**Rating:** 5
**Confidence:** 3

**Summary:**

The paper proposes the Memory Metropolis (MeMe) algorithm, integrating neural networks with simulated annealing (SA) to optimize combinatorial problems on 2D binary grids. By leveraging a unique class of network architecture termed "template networks," the method directs convergence towards states of structurally clustered patterns. This approach challenges conventional practices by intentionally violating the Markov property and is applied to nanophotonic inverse design, highlighting its potential in finding clustered design patterns.

**Strengths:**

* The introduction of "template networks" and the Memory Metropolis approach presents a fresh perspective in the realm of optimization.

* Combining elements from Markov Chain Monte Carlo optimization, neural networks, and reinforcement learning is interesting.

**Weaknesses:**

* The technical contribution is not strong. The proposal is generally mired in complexity which may make it inaccessible for readers not deeply familiar with all the integrated disciplines. Intentionally violating the Markov property without substantial justification is concerning. Further evidence or theoretical underpinnings are needed to support this decision.

* Rewriting for clarity can make the paper more accessible to a broader audience.

* The method of reward maximization and the process of determining detrimental actions is not explained in depth.

* Abstract is too lengthy

* The conclusion should reiterate the major findings, their implications, and potential future work in a more detailed manner.

**Questions:**

* A detailed side-by-side comparison with existing SA and regularized SA methodologies is required.

* Delving deeper into the reasons for violating the Markov property and the potential implications can make the proposal more convincing.

* Authors are suggested to discuss the broader applicability of the MeMe algorithm, beyond the specific case study presented.

* The paper doesn't clarify whether the approach is generalizable outside of the specific domain it was applied to.

---

> ### Author Response · Authors · 2023-11-17
>
> Thanks for your feedback.
> We updated the manuscript regarding abstract, methods and conclusion and by this made it more clear and accessible.
> By this we incorporated your suggestions listed under _Weaknesses_, except (1) which we are answering as part of your questions:
>
> 1. Since we do not only seek to maximize our devices fitness $f$, but also minimize the granularity $g$ we can not formulate a single objective which can be compared easily e.g. in a table, unfortunately.
>   Instead, we chose to depict both objectives in Fig. 4 for a direct side-by-side comparison.
>   In Fig. 4A SA as well as a pure policy gradient RL approach are given by the border cases of $T_S=\infty$ and $T_M=\infty$, respectively.
>   As indicated, regularized SA is a suitable alternative approach to MeMe, we thus include a comparison in Fig. 4B and further show, that MeMe outperforms regularized SA for any regularization factor $\lambda$ in Sec. A.3.
> 2. We agree, that the markov property should not be violated if one seeks to directly sample from the underlying distribution (here i.e., the Boltzmann distribution of fitness values).
>   However, we are only interest in finding a final state of high fitness while clustered structures are preferred.
>   Thus, violating the Markov property is not of big concern in our case which we state now more clearly in our text.\
>   Furthermore, we added a detailed analysis of the consequences of the violation of the Markov property in Sec. A.15.
>   We there conclude, that the violation actually results in a new Markov process, which generates samples from the product of the Boltzmann distribution of fitness values and the bias term $B = \exp{2Q/T_S}$ where Q is the learned template after the template matching operation (TMO).
>   We think this theoretical analysis substantially strengthen our motivation for violating the Markov property and hope this answers the question on the implications.
> 3. / 4. While we focused on an in-depth analysis of the problem of nanophotonic inverse design here, many other applications exist where similar structures are sought.
>   This is especially true for engineering where optimization of structures for various objectives and under specific constraints exist.
>   However, while many of these applications exist, they are ofentimes rather specific and require a detailed problem introduction as we did for the nanophotonic inverse design.

---

> > ### Comment · Reviewer_9yfn · 2023-11-23
> >
> > I appreciate the authors' efforts in addressing my previous concerns. However, I still have reservations regarding the technical contribution of this work, particularly in light of the absence of sufficient baseline comparisons. It is essential to establish the novelty and effectiveness of the proposed approach by comparing it with more (advanced) existing methods, and the absence of such comparisons is a notable limitation.
> >
> > Additionally and importantly (as far as I am concerned), the potential applicability of this work to a broader audience at ICLR raises concerns about its impact and relevance to the ICLR venue.
> >
> > Given these remaining doubts about the overall contribution of the paper, I have decided to revise my score only to '5: marginally below the acceptance threshold.'

---

### Official Review · Reviewer_SEFL · 2023-11-02

**Soundness:** 3 good
**Presentation:** 3 good
**Contribution:** 2 fair
**Rating:** 5
**Confidence:** 3

**Summary:**

This paper proposes a novel MCMC algorithm Memory Metropolis (MeMe), to tackle the combinatorial optimization problem in the context nanophotonic inverse design. The problem involves finding specially constrained patterns on a binary grid, with applications in creating high-performance devices for nanophotonic integrated circuits. MeMe involves the use of a neural network to build transition proposal distributions in Simulated Annealing (SA). The key contribution is 'template networks', a new class of network architectures designed to learn a template for constructing a proposal distribution for state transitions. MeMe violates the Markovian property as it uses past states to craft transition proposals. The template network is trained on the evaluation results of intermediate states of a single optimization run, which results in an architecture that does not require an input layer. Additional inductive biases are incorporated in the form of layers with limited local connectivity, which encourages the emergence of structural clusters. This biases the target distribution towards cluster formation. MeMe is also linked to deep RL, where the optimization objective of the Metropolis algorithm is viewed as a reward maximization problem. The policy is constructed using the discrepancy between the template and the current state, allowing the template network to find high-reward template-patterns. MeMe is evaluated empirically via application to combinatorial optimization in nanophotonic inverse design where it demonstrates significant improvements over standard SA.

**Strengths:**

* The paper studies the interesting problem appearing in the context of nanophotonic inverse design. The problem is described and formalized clearly, well motivated and presents a unique interesting challenge for machine learning approaches. This isn't the first instantiation of using sampling approaches for combinatorial optimization but is quite well executed.
* MeMe leverages advances from deep learning in the form of the template networks to craft effective proposal distributions within simulated annealing to model the biased target distribution to get high scoring candidates.
* The experiments on the nanophotonic design task is described in ample detail and thoroughly analysed.

**Weaknesses:**

* A major weakness in my opinion is that it is unclear how much of the method is generally applicable to other problem settings. It appears that the design of the template networks requires quite a bit careful engineering and domain knowledge and can be potentially challenging on other tasks. The paper's narrow focus on a specific application also makes it somewhat poorly positioned for the audience at ICLR, even though the domain is introduced appropriately. I encourage the authors to consider alternative venues where the particular application is a focus.
* Another major shortcoming is the lack of baselines - the authors only compare the apporach to simulated annealing but it would be good to have other baselines for instance some standard RL methods like PPO.

**Questions:**

* Can you provide more details on the computational cost of MeMe? How does it scale with the problem size?
* There is a potential issue of overfitting in the training of the template network? If so, how is it addressed?

---

> ### Author Response · Authors · 2023-11-17
>
> ***Weaknesses***
> 1. We agree that our application is focused on nanophotonic inverse design.
>   However, while other applications exist, they are oftentimes rather specific and require a detailed problem introduction as we did for the nanophotonic inverse design in our work.\
>   Furthermore, we think our proposed algorithm might be of interest to the ICLR audience from a conceptional perspective, which can only hardly be adresses in a more application focused venue.
>   The concept of input-less template networks is, to the best of our knowledge, new to deep RL in general and offers new perspectives especially for optimization applications.
> 2. We agree that comparisons to additional algorithms would be interesting, however, to the best of our knowledge there are no promising alternative algorithms for the problem we studied in our work.
>   When analyzing the effect of the sampling temperature $T_S$ and the metropolis temperature $T_M$ in detail, multiple baseline algorithms emerge in their borderline cases.
>   Next to SA and greedy-exploration, this also includes a direct policy gradient apporach if $T_M=\infty$.
>   Furthermore, we also use loss clipping, which is a key part of PPO.
>   As training our template networks differs from classical deep RL in many aspects, we many components which where developed for other RL agents like PPO, are missing motivation for our proposed algorithm.
>   This includes value functions, lower discounts, rollouts of multiple samples, multiple policy update steps (as in PPO objective maximization).
>   Instead, we seek to update the learned template as direct and fast as possible with newly encountered feedback from the environment.
>   We thus included features of e.g. PPO which are transferable to our application while think that more sophisticated RL methods are not well motivated as a baseline for our application.
>
>
> ***Questions***
> 1. The computational cost of our algorithm is dominated by the FDFD-simulations.
>   So an empirical evaluation would only observe the scaling of these (which scales quadratic with the system size).
>   E.g. for our experiments in Fig. 4 computations took $\approx 20$h on 26 Intel Xeon Gold 6140 2.30 GHz CPU cores per optimization.\
>   Since updates of the template network are only propagated to a limited local field of view, training as well as updates of the proposal distribution are scaling linear with the number of pixels (instead of quadratic as expected for e.g. FC layers).
> 2. Indeed, we intend to fully fit our networks to single optimization runs as a key concept of template networks.
>   This distinguises template networks from most other architectures used in deep RL.
>   In this regard out networks share a lot with e.g. NeRFs which are also fitted to single scenes.
>   Thus overfitting is not of concern.

---

> > ### Comment · Reviewer_SEFL · 2023-11-21
> > **Response to Rebuttal**
> >
> > Thanks for the response and clarification!
> >
> > > The concept of input-less template networks is, to the best of our knowledge, new to deep RL in general and offers new perspectives especially for optimization applications.
> >
> > It would be very helpful for me if the authors could discuss some other potential applications where the template networks could be applicable. That remains the primary concern I have about the paper.

---

### Official Review · Reviewer_UWqi · 2023-11-08

**Soundness:** 3 good
**Presentation:** 2 fair
**Contribution:** 2 fair
**Rating:** 5
**Confidence:** 2

**Summary:**

The paper suggests a combination of neural network-based approach and deep RL to the problem of combinatorial optimization with the simulated annealing algorithm. The proposed algorithm utilizes the RL approach to construct the proposal particles in the modification of Metropolis-Hastings scheme. The authors also provide a results on physical simulations demonstrating the efficiency of their approach compared to the vanilla simulated annealing scheme.

**Strengths:**

The topic of combining RL with discrete optimization is challenging, and the experimental results of the submission are spectacular, especially in the term of quite large problem dimension.

**Weaknesses:**

The relation of the proposed algorithm to the RL setting is not clearly explained in the current submission. Current submission lacks the detailed MDP description with the tuple of state space, action space, and reward, and the reward description for the particular optimization problem. The writing of section 3, and especially section 3.1 is hard to follow. The choice of extremely discounted RL problem (with $\gamma = 0$) is also rather questionable for an empirical paper, and requires additional experimental verification. Moreover, there already were papers, e.g. [Beloborodov et al, 2020], [Mills et al, 2020], which already provided a framework for treating SA as an MDP and applied RL for solving it. That is why, I suggest the authors to better indicate the novelty of their approach.

[Beloborodov et al, 2020] Beloborodov, D., Ulanov, A. E., Foerster, J. N., Whiteson, S., & Lvovsky, A. I. (2020). Reinforcement learning enhanced quantum-inspired algorithm for combinatorial optimization. Machine Learning: Science and Technology, 2(2), 025009.
[Mills et al, 2020] Mills, Kyle, Pooya Ronagh, and Isaac Tamblyn. "Finding the ground state of spin Hamiltonians with reinforcement learning." Nature Machine Intelligence 2.9 (2020): 509-517.

**Questions:**

I would suggest the authors to add more structure to the current version of section 3, adding more details on how the considered problem falls into the RL formalism.

Moreover, I would like the authors to elaborate the novelty of their suggested algorithm. For example, RL approach to simulated annealing  was recently considered, e.g. in [Correia et al, 2023], and references therein. Thus I would suggest the authors to better highlight the novelty of their approach compared to the ones discussed in the previous papers.

References:
[Correia et al, 2023] Correia, Alvaro HC, Daniel E. Worrall, and Roberto Bondesan. "Neural simulated annealing." International Conference on Artificial Intelligence and Statistics. PMLR, 2023.
[Beloborodov et al, 2020] Beloborodov, D., Ulanov, A. E., Foerster, J. N., Whiteson, S., & Lvovsky, A. I. (2020). Reinforcement learning enhanced quantum-inspired algorithm for combinatorial optimization. Machine Learning: Science and Technology, 2(2), 025009.
[Mills et al, 2020] Mills, Kyle, Pooya Ronagh, and Isaac Tamblyn. "Finding the ground state of spin Hamiltonians with reinforcement learning." Nature Machine Intelligence 2.9 (2020): 509-517.

---

> ### Author Response · Authors · 2023-11-17
>
> Thanks for your feedback and suggesting the additional references.\
> We added the references to our related work.
> Correia et al. indeed propose a similar approach by combining RL and SA to generate new state proposals.
> However, they do not use the learned proposal distribution to deliberately bias the optimization to converge to a constrained state.
> Actually they also do not apply a sampling bias correction (as discussed in Sec. A.9) resulting in uncontrolled biases of converged states, which might offer an explanation for their comparably bad results to SOTA algorithms.
> Furthermore, they rely on hand-crafted features as inputs to FC layers to parameterize the proposal distribution, where we propose input-less template networks, which is a central contribution of our work which we now indicate more directly in the main text.
> While we show that a simple policy gradient algorithm can be used to effectively train the template network, Correia et al. have to fall back to less interpretable evolution strategies to train their fully connected networks to reach sufficient results in many of their experiments.\
> We restructured and updated section 3, including a precise definition of the MDP accordingly, and added Sec. A.16 where we summarize our notation, thanks for the helpful suggestions.

---

### Meta-Review · Area_Chair_TCdE · 2023-12-07

**Metareview:**

This paper considers combinatorial optimization with an application in nanophotonic inverse design. It views the optimization objective as a reinforcement learning problem, introduces input-less template networks for simulated annealing and Memory Metropolis that violates the Markov property in the Metropolis algorithm intentionally.

Multiple reviewers consider the proposed method is well executed in the context of the specific application of nanophotonic inverse design. The idea of template networks and the Memory Metropolis are also new.

However, there is shared concerns on the scope of the paper. The proposed method is very specifically designed for its application domain. Reviewers are not sure if these methods have a boarder application or can be of interest for a wider ML community. The authors discussed other potential applications in their rebuttal but it did not solve the concerns. Besides, multiple reviewers would also like to see comparison with related approaches, e.g. the references from reviewer UWqi. This submission might be more suitable for an application related venue. To prepare for a publication at ICLR, I would recommend the authors to explain the problem and their method in a more general problem setup, compare with most related methods, and evaluate its efficacy in the nanophotonic inverse design problem.

**Justification For Why Not Higher Score:**

The concern on the scope of the proposed method is not resolved in the rebuttal.

**Justification For Why Not Lower Score:**

N/A

---

### Decision · Program_Chairs · 2024-01-16

Reject